# Urban development pattern's influence on extreme rainfall occurrences

Long Yang [1,2] ✉, Yixin Yang[1,2], Ye Shen[1,2], Jiachuan Yang[3], Guang Zheng[4], James Smith[5] & Dev Niyogi[6,7]

Growing urban population and the distinct strategies to accommodate them lead to diverse urban development patterns worldwide. While local evidence suggests the presence of urban signatures in rainfall anomalies, there is limited understanding of how rainfall responds to divergent urban development patterns worldwide. Here we unveil a divergence in the exposure to extreme rainfall for 1790 inland cities globally, attributable to their respective urban development patterns. Cities that experience compact development tend to witness larger increases in extreme rainfall frequency over downtown than their rural surroundings, while the anomalies in extreme rainfall frequency diminish for cities with dispersed development. Convection-permitting simulations further suggest compact urban footprints lead to more pronounced urban-rural thermal contrasts and aerodynamic disturbances. This is directly responsible for the divergent rainfall responses to urban development patterns. Our analyses offer significant insights pertaining to the priorities and potential of city-level efforts to mitigate the emerging climate-related hazards, particularly for countries experiencing rapid urbanization.

Rapid growth of urban population stimulates significant urban expansions across the globe[1], resulting in multisector environmental changes that extend beyond urban boundaries[2]. One of the prominent effects of urban expansion is the alternation of thermodynamic (e.g., albedo, specific heat capacity) and aerodynamic (e.g., surface roughness) properties of land surface, along with concentrations of microphysical ingredients (e.g., anthropogenic aerosols) for cloud formation[3]. These modifications exert distinct influences on rainfall patterns both spatially and temporally[4], and subsequently, the potential of water-related hazards such as flooding over urban areas[5].

Predictive understanding of urban-induced rainfall anomalies, in terms of both their magnitudes and positions, remain challenging[6]. This is mainly due to the variant nature of cities, including size[7], shape[8], and geographical context (e.g., in the vicinity of land/water boundaries and complex terrain)[9,10], as well as the intricate dependence of rainfall variability on synoptic forcings[11], background climate[12,13], and microphysical properties of urban aerosols[14]. In addition, cities vary in how urban elements such as buildings and roads are spatially organized in horizontal dimension, referred to as urban footprints. The spatial disparity of urban footprints has become notable due to the emerging urban agglomerations worldwide but has not been considered in comprehending urban-induced rainfall anomalies.

Contrasting urban footprints are the result of different urban development histories and strategies. For instance, compact urban development pattern, premised on efficient land use, leads to urban intensification and the aggregation of urban elements into a single cluster. By contrast, urban footprint from dispersed urban development pattern, also known as "urban sprawl", comprises scattered

[1]School of Geography and Ocean Sciences, Nanjing University, Nanjing, China. [2]Frontiers Science Center for Critical Earth Material Cycling, Nanjing University, Nanjing, China. [3]Department of Civil and Environmental Engineering, The Hong Kong University of Science and Technology, Hong Kong, China. [4]International Institute for Earth System Science, Nanjing University, Nanjing, China. [5]Department of Civil and Environmental Engineering, Princeton University, Princeton, NJ, USA. [6]Department of Geological Sciences, Jackson School of Geosciences, The University of Texas at Austin, Texas, TX, USA. [7]Department of Civil, Architectural, and Environmental Engineering, Cockrell School of Engineering, The University of Texas at Austin, Texas, TX, USA. ✉ e-mail: yanglong@nju.edu.cn

residential regions and the disaggregation of urban elements into multiple clusters[15]. The synergies between different urban development patterns and environmental (e.g., energy consumption, $CO_2$ emissions)[16,17] and social sustainability (e.g., subjective well-being)[18] have received considerable attention. However, much less is known regarding the role of different urban development patterns in altering extreme rainfall and the resultant flood risks, especially with representations of global cities at various development levels. This may undermine several global initiatives (e.g., the Urban Climate Change Research Network)[19,20] aimed at sustainable urban development, particularly considering that urban regions are particularly vulnerable to extreme rainfall and flooding under a changing climate[21,22].

Here we reveal contrasting urban development patterns (in terms of areal changes in urban coverage and the spatial aggregation of urban elements) for 1790 inland cities (i.e., free of land-water boundaries and complex terrain) globally during the period 2003-2018 (see Methods). Cities in high- and upper middle-income countries (according to the World Bank classification system by income level for FY24, same below) tend to experience compact urban development. Cities in low-income countries show faster expansion in impervious areas, but this expansion tends to be characterized by dispersed urban development. The magnitudes and spatial patterns of rainfall changes varied across city groups with different urban development patterns. Convection-permitting modeling analysis shows that spatially aggregated urban footprints (i.e., compact cities) pose strong thermodynamic and aerodynamic disturbances to synoptic forcings. These disturbances are responsible for more pronounced changes in extreme rainfall occurrence and increased rainfall accumulation within the urban boundaries of compact cities than in their surrounding rural areas. Weak urban-rural contrast in rainfall anomalies, however, imply extended exposure to flood hazards for both urban and rural residents under dispersed urban development. Our analyses provide notable evidence to urban planners and policymakers for prioritizing measures in sustainable urban development, particularly for countries experiencing rapid urbanization.

## Results

### Global urban development patterns

From 2003 to 2018, all 1790 cities experienced expansion in urban coverage; however, they exhibit distinct development patterns in terms of the magnitude of areal expansion and the spatial aggregation of urban elements (Fig. 1). Here we only consider inland cities without complex terrain or notable land-water boundaries so that urban development pattern is less likely to be constrained by their physiographic environments. We focus on a $1° × 1°$ domain, with its center aligned to the geographic coordinates of each respective city. The choice of $1° × 1°$ domain is the compromise of high-resolution data availability and the practicality of encompassing most metropolitan regions worldwide in our analysis. When cities are small in size, the $1° × 1°$ domain represents the aggregation patten of several urban clusters.

The 1790 cities can be categorized into three groups according to their development patterns (see Methods). Cities in Group I ($N = 1105$) are characterized by relatively low urban ratios (with a mean value of 5%, i.e., the percentage of urban area in the $1° × 1°$ domain) and low urbanization rates (with a mean value of 7%, i.e., relative changes in urban coverage). These cities exhibit a broad spatial distribution across the globe. Although cities in Group III ($N = 268$) are characterized by comparable urban ratios (with a mean value of 4%), they demonstrate pronounced urbanization rates (with a mean value of 27%, Supplementary Fig. 1). Urban elements tend to be developed in a more spatially disaggregated way, i.e., a dispersed urban development

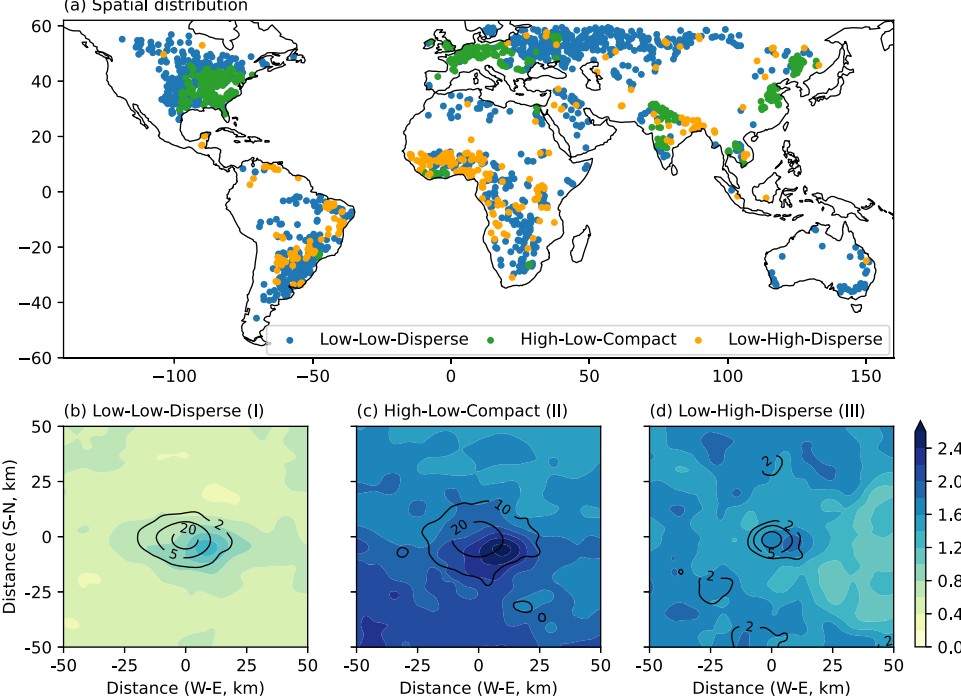

**Fig. 1 | Divergent urban development patterns and the associated anomalies in spatial rainfall patterns. a** Spatial distribution of cities with three different development patterns; (**b**–**d**) composite mean change ratios in extreme rainfall frequencies (i.e., exceeding the 99th percentile daily rainfall of rainy days, represented by the shade) for different city groups with diverse development patterns between the period 2000–2005 and 2016–2020. **b** Group I (include cities with low urban coverage), low increases in urban area and slight increase in landscape shape index, i.e., "Low-Low-Disperse" cluster, (**c**) Group II (include cities with high urban coverage, low increases in urban area and slight decreases in landscape shape index, i.e., "High-Low-Compact" cluster, (**d**) Group III (include cities with low urban coverage, high increases in urban area and notable increases in landscape shape index, i.e., "Low-High-Disperse" cluster). The contour shows the normalized number of urban pixels (i.e., by diving the maxima within the domain, in percentage), providing an approximation of the city boundary.

pattern. This is evidenced by the increased landscape shape index (an urban morphological metric characterizing the extent of aggregation of urban elements, see *Methods*). These cities are mainly situated in low- and middle-income regions, including southeastern South America, western Africa, and northeastern India (Fig. 1).

Cities in Group II ($N = 417$) exhibit the highest urban ratio (mean value of 43%) among the three groups, while maintaining similar urbanization rates (mean value of 6.8%) to Group I (Supplementary Fig. 1). These cities tend to adopt a compact urban development pattern, with new urban elements more spatially aggregated towards existing urban coverage, compared to the other two city groups. These cities are clustered in upper middle/high-income countries or densely populated regions, including eastern US (i.e., east of the Rocky Mountains), western Europe, south Asia, and northern China (Fig. 1). Analysis of urban development patterns highlights globally distinct strategies to accommodate the growing urban population over the past two decades.

## Contrast urban signatures in rainfall anomaly

Diverse urban development patterns further lead to varying urban signatures in rainfall anomalies. Composite analyses show that expansions in urban coverage overall contributes to increased frequencies of extreme rainfall (i.e., exceeding 99th percentile daily rain rate over all rainy days) over global cities (see "Methods"). The most significant increase in extreme rainfall frequency is observed near the city center, gradually decreasing as one moves outward (Fig. 1). The pronounced gradients of rainfall anomalies highlight urban influences, as we anticipate that climate effects will exhibit consistent behavior across the entire city domain. This finding echoes urban rainfall anomalies revealed from previous continental-scale studies[23–25].

Changes in rainfall patterns show notable variations among the three city groups. The maximum change ratio of extreme rainfall frequency is 1.9% for Group III, only slightly larger than 1.5% for Group I. However, the domain-average change ratio for Group III (i.e., 1.5%) is three times as large as that for Group I (i.e., 0.5%). This corresponds to approximately three times larger urbanization rates for Group III, even though the two groups have similar types of urban footprints (Supplementary Fig. 1). A notable feature is that there are multiple "hotspots" of large rainfall increments across the city domain for Group III (Fig. 1). This pattern could potentially be linked to the inclination of urban development to occur in a spatially disaggregated manner. The establishment of newly developed residential or commercial clusters can result in heightened frequencies of extreme rainfall in the vicinity.

Both the maximum (i.e., 2.7%) and domain-average (i.e., 1.8%) changes in extreme rainfall frequency are notably larger for Group II than Group I. This is because Group II exhibits a larger spatial coverage of built-up area, thus a larger percentage of transitions from "nonurban" to urban pixels are required to sustain similar urbanization rates as seen in Group I. These new urban pixels tend to aggregate to the existing urban patches (i.e., a compact urban development). This contributes to increased occurrences of extreme rainfall within cities, contrasting with the dispersed "hotspots" observed in Group III. The rainfall contrasts among the three city groups persist using the 90th percentile of daily rainfall as the extreme rainfall threshold (Supplementary Fig. 2). The contrasting rainfall patterns across three city groups persist in different climate zones, indicating little impacts of background climate on divergent rainfall responses to different urban development patterns (see Methods and Supplementary Fig. 3).

## Impacts of urban footprint on rainfall

To understand the impacts of different urban development patterns on rainfall, we carry out a series of numerical simulations based on the Real Atmosphere, Ideal Land surface (RAIL) approach (see Methods). We configure five urban scenarios (i.e., Circular, Ribbon, Satellite, Edge, and Scatter, Supplementary Fig. 4) and one "no-city" scenario

(i.e., only cropland). The configurations of urban scenarios are modeled after real-world cities, so that different urban footprints can be realistically represented in numerical simulations (see Methods). Circular and Ribbon scenarios represent compact cities (i.e., outcomes of compact urban development), while Satellite, Edge, and Scatter scenarios represent dispersed cities (i.e., outcomes of dispersed urban development).

The spatial distributions of rainfall accumulation are similar across various scenarios, suggesting that there is limited disturbance from urban canopy processes on synoptic forcings (Supplementary Fig. 5). Distinct urban rainfall signatures become evident throughout the city domain. The presence of urban coverage leads to decreased rainfall accumulation over the city domain, but the extent of this reduction varies among different urban scenarios, approximately 0.8 mm–2.6 mm (Supplementary Fig. 6), that is 3–10% (i.e., the mean rainfall over the urban domain is about 25 mm). Specifically, in the Circular city and Ribbon city scenarios, there is a statistically significant reduction in domain-average rainfall of $-2.3 \pm 0.65$ mm ($P < 0.01$) and $-2.6 \pm 0.69$ mm ($P < 0.01$), respectively (Supplementary Fig. 6). For the other three urban scenarios, changes in domain-average rainfall are not statistically significant. It is worth mentioning, however, that total rainfall is increased over the urban grids in the Circular city ($0.8 \pm 0.3$ mm, $P < 0.05$) and Ribbon city scenarios ($3.5 \pm 1.2$ mm, $P < 0.05$, Supplementary Fig. 7).

In contrast to the spatial anomalies in rainfall accumulation, we observe a rise in the frequency of extreme rainfall occurrences (i.e., measured by the total number of hourly rain rates exceeding 10 mm/h) in all urban scenarios. The increases are particularly pronounced in the two compact city scenarios, i.e., Circular and Ribbon, with distinct "hotspots" of elevated occurrences (~2 h) emerging along the urban-rural interface (Fig. 2). The spatial pattern persists using 2 mm/h as the threshold for extreme rainfall (Supplementary Fig. 8). Elevated occurrences of extreme rainfall are more spatially disaggregated and tend to spread across the domain for the Satellite, Edge, and Scatter city scenario (Supplementary Fig. 9). This is consistent with how the urban elements are spatially organized for the three urban scenarios. More frequent extreme rainfall is directly responsible for increased rainfall accumulation over urban grids in the two compact city scenarios (i.e., Circular and Ribbon, Supplementary Fig. 7).

To further highlight contrast urban signatures in spatial rainfall anomalies, we carry out buffer analysis (see "Methods"). Urban signatures in extreme rainfall occurrences rapidly fade for the two compact city scenarios (i.e., Circular and Ribbon) as the buffer distance is gradually increased outward (Fig. 2f). The increase in extreme rainfall frequency over the two compact city scenarios (Circular and Ribbon, i.e., 0.30), relative to the "no-city" scenario, is approximately three times faster than that in the three dispersed cities, i.e., 0.10 (Fig. 2f). The rainfall anomalies induced by urban footprints become inconspicuous when the buffer region size is approximately five times as large as the urban footprint (i.e., city size). A similar rule applies for total rainfall (Supplementary Fig. 10). The contrasts across different scenarios become insignificant when the buffer region extends to the entire model domain, due to the dominance of synoptic forcing in rainfall at regional scales. These analyses suggest that compact urban development (i.e., Circular and Ribbon) tends to amplify rainfall accumulation and extreme rainfall occurrences over cities, whereas dispersed urban development (i.e., Satellite, Edge, and Scatter) may lead to more scattered disturbances in rainfall distribution, as also reflected in the satellite-retrieved rainfall anomalies for cities that experience fast urbanization (Fig. 1).

Contrasting urban signatures in rainfall anomalies are intricately linked to the thermodynamic and dynamic disturbances caused by different urban footprints (Supplementary Fig. 11). These disturbances play a critical role in rainfall processes. Compared to the two compact city scenarios (i.e., Circular and Ribbon), the three dispersed city

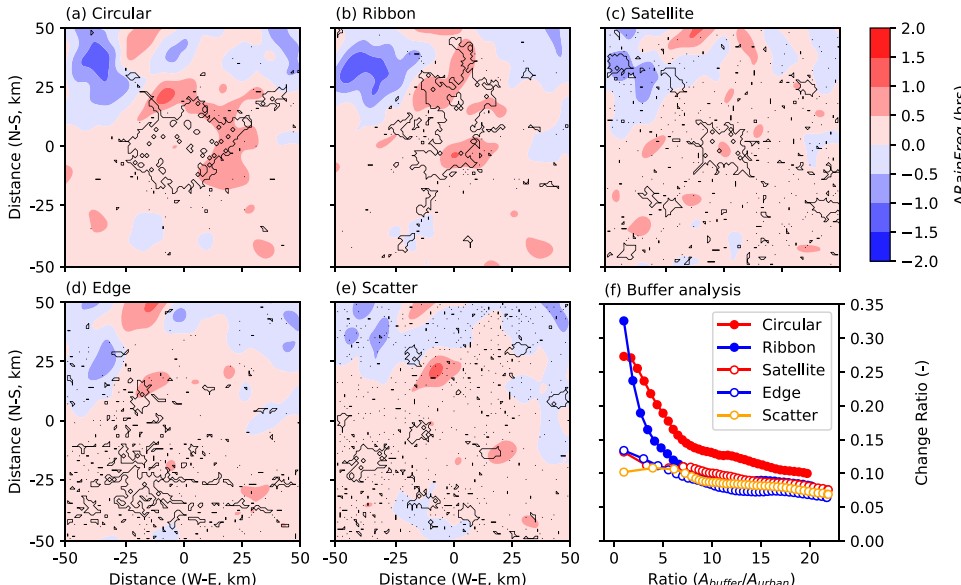

**Fig. 2 | Spatial pattern of differences in extreme rainfall occurrences (represented by number of hours with rain rate exceeding 10 mm/h) between each urban scenario and the "no-city" scenario. a** Circular, (**b**) Ribbon, (**c**) Satellite, (**d**) Edge, (**e**) Scatter. The averaged change ratio, i.e., represented as $(R_{urban}-R_{no\text{-}city})/(R_{urban}+R_{no\text{-}city})$, within urban boundary and its buffering regions is shown in (**f**). The contours in (**a**)–(**e**) highlight the urban footprint for each scenario.

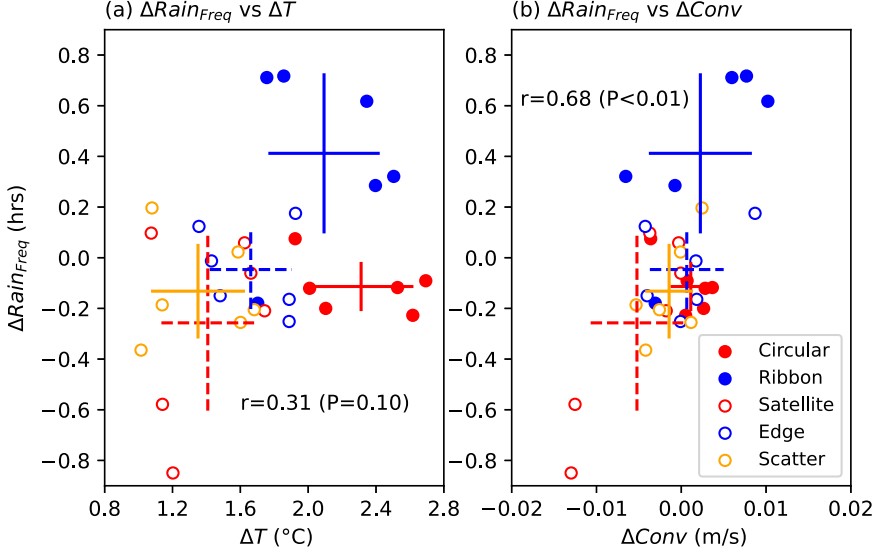

**Fig. 3 | Relationship between changes in extreme rainfall occurrences within the urban boundary ($\Delta Rain_{Freq}$, in hours) and responsible variables. a** 2-m temperature ($\Delta T$, in °C) averaged over the pre-storm period; (**b**) vertical velocity at 850 hPa ($\Delta Conv$, in m/s) averaged during the storm period. Each circle represents a single ensemble member of the corresponding urban scenario. The intersected lines show the standard error for each urban scenario. Pearson correlation coefficients between the variables and their corresponding statistical significance are shown in each subplot title.

scenarios (i.e., Satellite, Edge, and Scatter) show slightly larger surface temperature anomalies over the domain, approximately 0.32 °C compared to 0.29 °C (Supplementary Fig. 12). This is possibly tied to the enhanced advection of sensible heat from cities to neighboring rural regions, as urban footprints are more widely distributed across the domain. However, the urban-rural thermal contrast, also known as urban heat island (UHI) intensity, is approximately 0.5 °C higher for the Circular and Ribbon city scenarios (mean values of 1.4 and 1.2 °C, respectively) than that for the Satellite (mean value of 0.8 °C), Edge (mean value of 0.9 °C), and Scatter (mean value of 0.7 °C) city scenarios (Supplementary Fig. 12). This is due to sharper contrasts in aerodynamic resistance between compact urban grids and the surrounding cropland[26]. The enhanced urban-rural thermal contrast, and thus surface pressure gradients subsequently facilitate low-level moist convergence into cities, represented by vertical velocity at the level of 850 hPa (Supplementary Fig. 13). Anomalies in extreme rainfall occurrences over urban grids thus exhibit a correlation with UHI intensity (r = 0.31, *P* = 0.10), and demonstrate a statistically significant correlation with changes in low-level convergence (r = 0.68, *P* < 0.01) during the storm period, despite variations across different ensemble members (Fig. 3).

The increased aerodynamic roughness (as represented by changes in friction velocity) corresponds well with changes in low-level convergence (r = 0.51, *P* < 0.01, Supplementary Fig. 14). Compared to

the three dispersed city scenarios, surface roughness for the two compact city scenarios is larger. This is further confirmed by the agreement between the spatial distribution of enhanced low-level convergence and the "hotspots" of elevated extreme rainfall occurrences in the Circular and Ribbon city scenarios (Supplementary Fig. 13). These analyses emphasize the significance of interactions between thermodynamic and dynamic disturbances caused by various layouts of urban elements in influencing extreme rainfall processes.

## Discussion

Figure 4 summarizes the conceptual understanding pertaining to contrasting rainfall responses to different urban development patterns. Contrasting impacts of different urban development pattern on rainfall have important implications for sustainable urban planning and city-level actions for the emerging climate-related hazards. This significance is particularly pronounced for many worldwide regions that are undergoing or projected to experience notable urban expansions. Compared to most high-income regions, such as eastern US, western Europe, and northeastern China, low- and middle-income countries in Africa, South America, and South Asia are experiencing larger urbanization rates and tendencies of dispersed urban development. This may offer relief from the impacts of enhanced extreme rainfall over downtown, but inevitably enhance the exposure of rural residents to rainfall-related hazards. It is important to note that flood defense facilities are typically weaker in rural areas compared to downtown regions. Escalated investments are thus needed for improved monitoring and forecasting capabilities across the entire urban agglomeration region. Upper middle- and high-income regions

that experience compact urban development need to cope with the unintended convergence of increased frequency of extreme rainfall events and concentration of assets.

A caveat of our analysis is that we use a 1° × 1° domain to characterize urban development patterns. We thus note that our results offer aggregated characterization of large urban regions globally and their interactions with rainfall, but may be less applicable for individual small cities. The choice of a coarse domain is mainly constrained by the limited availability of globally, high-resolution rainfall dataset. The 1-km MRMS radar rainfall products across the continental United States, can be further utilized for regional analysis. This enables characterization of urban development patterns in a finer spatial scale. The emerging satellite-based products of global building footprint products can additionally provide valuable three-dimensional characterization of global cities as well as their temporal evolution patterns[27]. Unveiling the relationship between three-dimensional urban development patterns and regional climate is worthy to be investigated in future studies. The climate resiliency of urban areas can be accordingly enhanced by fully characterizing how cities develop in time (as represented by temporal changes in spatial coverage, density, aggregation pattern, etc.) as well as the interactions of these changing patterns with regional climate.

Our analyses highlight the importance of fine-scale characterization of heterogeneous land surface properties in understanding the interactions between urban canopy processes and the lower atmosphere. The representation is either absent in most state-of-art global climate models, or based on bulk parametrizations derived from a mosaic approach where surface scalar and momentum fluxes are calculated by simply weighting the contributions over urban and non-urban sub-grid pixels[28]. The resultant bias over urban areas can potentially influence their rural neighbors through atmospheric "chain" flows and would be likely expected for rapidly urbanizing regions that are characterized by diverse strategies of urban development. Our results call for updates of current city-level projections of climate change impacts[29,30]. This emphasizes the utility of high-resolution regional climate simulations incorporated with improved urban parametrization schemes[31] and advanced geographic products that accurately characterize heterogeneous land surface properties (including urban fabrics, anthropogenic activities, etc.)[32,33]. It is also important to develop tools that can efficiently assess the amplified climate effect linked to different urban development strategies, so that urban planners and policymakers can be better informed for sustainable urban planning.

## Methods
### Urban development analysis
We analyze global urban development patterns based on the 30-m resolution Global Artificial Impervious Areas (GAIA) dataset. GAIA has been evaluated against other global urban products, and demonstrates good performance[34]. We aggregate and resample the 30-m product into a resolution of 0.01° using the Python package GDAL (i.e., consistent with rainfall product used in the following analyses, see details below). The value of each 0.01° grid is assigned by the total number of 30-m impervious pixels (i.e., 30-m) aggregated into it. We define the 0.01° grid as an urban grid when the impervious ratio, i.e., total number of impervious pixels divided by total pixels aggregated into a 0.01° grid, exceeds 0.2 (i.e., similarly adopted by some other land use datasets, e.g., NLCD, in classifying urban grids). This makes our analyses less contaminated by variant changes in sparse urban units (e.g., parks, golf courses, etc.) that are not for residential or commercial/industrial purposes.

We examine urban development in the past two decades by comparing differences in urban coverage between the periods 2001–2005 and 2016–2020. The urban layers from 2003 and 2018 are used to represent the mean urbanization status during the two distinct periods, respectively. Our analysis is carried out from a city-centric

(a) Compact city

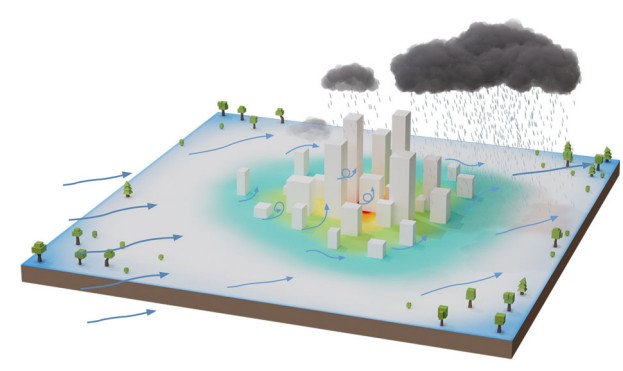

(b) Dispersed city

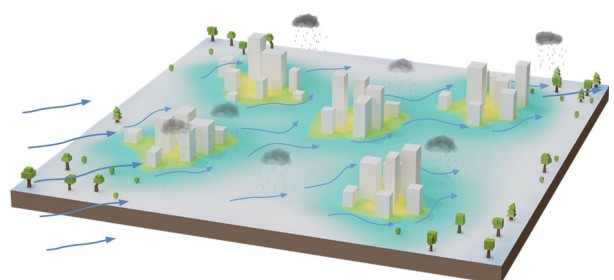

**Fig. 4 | The conceptual model for divergent rainfall responses to contrasting urban development patterns. a** Compact city scenario, (**b**) dispersed city scenario. The shade on the ground represents the urban-rural thermal contrast, with warm (cold) colors indicating high (low) surface temperatures. The vectors represent both the magnitude and direction of synoptic flows.

perspective. We obtain a list of global cities from the Natural Earth archive. We exclude cities from the list if they are close to coasts (with coastal lines included in their city-center domain) and/or mountainous terrain (maximum elevation difference exceeds 500 m). We analyze the elevation difference based on the SRTM30+ Global 1-km Digital Elevation Model (Version 11). A total of 1790 inland cities are selected for further analysis. We focus on a 1° × 1°domain, with its center aligned to the geographic coordinates of each respective city. We quantify urban development patterns based on two landscape metrics within each domain. The two metrics are total urban coverage (TA) and landscape shape index (LSI):

$$TA = \sum_{j}^{n} a_j \tag{1}$$

$$LSI = \left(0.25 \sum_{k=1}^{m} e_k\right) / \sqrt{TA} \tag{2}$$

where $a_j$ is the unit area of an urban grid; $n$ is the total number of urban grids; $e_k$ is the total length of edge between an urban grid and its non-urban neighbors; $m$ is the total number of patches in each city domain. Urban ratio is represented by TA divided by domain size. Changes in the two metrics between 2003 and 2018 shed light on how urban elements are changed in both area and spatial organizations. LSI is a useful index to represent the degree of aggregation in relation to shape complexity, especially when comparing a range of sizes[35,36]. For instance, larger LSI indicates that urban footprint is more spatially disaggregated. Increases (decreases) in LSI are then indicative of a dispersed (compact) urban development pattern.

There are other metrics that characterize degree of aggregation. We compare LSI with three other dimensionless metrics, i.e., number of patches, patch density, and splitting index (see Supplementary Table 1 for details). These metrics are significantly correlated with LSI. This is because fine-scale characterization of urban aggregation patterns can not be revealed by the coarse scale of analysis (i.e., 1°). We use LSI in the following analyses due to its capability of characterizing diverse urban development patterns. All the landscape metrics are calculated based on the Python package PyLandStats.

We employ the K-means clustering algorithm (ref. [37]) to categorize the 1790 cities into distinct groups according to their urban development patterns. We assess various combinations of descriptors for the clustering (Supplementary Table 2). We determine three as the optimal number of clusters. This choice is informed by the consistent attainment of the highest Silhouette score and Davies-Bouldin score (Refs. [38,39], Supplementary Fig. 15). The spatial distributions of three clusters are remarkably consistent with different combinations of descriptors. Group I include cities with low urban ratios, relatively low urbanization rates (i.e., changes in urban coverage), and demonstrate a slight tendency for dispersed urban development (i.e., changes in LSI). Group I is thus termed as the "low-low-disperse" cluster (N = 1105). Group II includes cities with high urban ratios, relatively low urbanization rates, and demonstrate tendencies for compact urban development, termed as the "high-low-compact" cluster (N = 417). Group III includes cities with low urban ratios but high urbanization rates, demonstrating notable tendencies for dispersed urban development, termed as the "low-high-disperse" cluster (N = 268).

We note that only two-dimensional characteristics of urban development patterns are considered here. This is because development of products that capture and characterize the three-dimensional morphology of global cities and their dynamics is in its early stages. For instance, characterizing three-dimension urban morphology of global cities during early 2000 s is still not feasible. Three-dimensional characterization of urban development should be a future endeavor in future studies.

## Rainfall analysis

We use the Integrated Multi-Satellite Retrievals of Global precipitation measurement (IMERG, Version 06) product. We only focus on precipitation in liquid form. The spatial and temporal resolutions are 0.1 degree and daily, respectively. We choose the IMERG product for rainfall analysis due to its quasi-global spatial coverage (60 °S - 60 °N). This enables us to represent changing rainfall patterns for high-latitude cities (e.g., northern Europe). We do not utilize reanalysis rainfall products due to their inadequate representation of intricate urban processes in current climate models. Global rainfall datasets based on interpolation or merging from rain gauges are neither used due to the incapability of gauges in capturing the variant urban rainfall signatures[40]. Continental or regional scale analyses with rainfall products of higher spatial resolutions, such as the Multi-Radar/Multi-Sensor precipitation reanalysis over United States (-1 km), can be pursued in future.

We calculate the total number of extreme rainfall days (RFreq99) for the two periods, i.e., 2001–2005 and 2016–2020, respectively. An extreme rainfall day is defined when the daily rain rate exceeds the 99th percentile over all rainy days (>0 mm/day) during each respective five-year period. We do sensitivity tests by using the 90th percentile as the threshold (RFreq90). The data layers of these rainfall statistics are consistent with the grid spacing of the aggregated urban product. Changes in rainfall statistics between two different periods are calculated as:

$$\Delta X = (X_{2016-2020} - X_{2001-2005}) / (X_{2016-2020} + X_{2001-2005}) \tag{3}$$

Where X can be replaced by different rainfall statistics, i.e., RFreq99, RFreq90. We composite changes in different rainfall statistics for cities of each clustering group over the 1° × 1° city-centered domain.

## Sensitivity analyses

We choose a 1° × 1° domain (approximately 0.01 million km²), because it enables us to fully cover most metropolitan areas in the world, as they cover less than 0.01 million km². The 1° × 1° domain also represents the best practice of existing global climate models in terms of spatial resolution. Using a smaller domain (e.g., 0.5° × 0.5°) can only include individual cities, and is not sufficient to represent diverse urban footprints. We also carry out analysis using a larger domain, i.e., 2° × 2°, with almost identical global patterns found and rainfall anomalies across the three city groups (Supplementary Fig. 16).

We test the sensitivity of the contrasting rainfall patterns to the uneven size of each group. We randomly choose 400 cities from Group I (i.e., comparable size to Group II), and derive the composite mean rainfall profile along the west-east and south-north direction, respectively. The random selection is done for 100 times, so that we have 100 composite profiles along each direction. We also test by choosing 200 cities from Group I (i.e., comparable size to Group III). Results show little variance to the rainfall profiles on the changes of group size (Supplementary Fig. 17).

To examine whether contrasting rainfall patterns across three city groups depend on background climates, we classify each city (according to their locations) into different climate types. We adopt four climate types (i.e., tropical, dry, temperate, continental) corresponding to the first level of the Köppen climate classification. There are no cities in polar climate type. The number of cities in each climate group is 395 (tropical), 384 (dry), 406 (temperate), and 606 (continental). For each climate type, we compare rainfall patterns with different urban development patterns. We find that the contrasting rainfall patterns across different city groups persist in different climate types (Supplementary Fig. 3). This may be partially due to the adoption of 1° × 1° domain which is insufficient to represent the development patterns of small cities, but mostly represent large metropolitan regions.

## Urban footprints

We examine the GAIA impervious product and select five cities as our prototypes of urban footprints. The five cities are Austin (US), San Antonio (US), Fuyang (China), Zhoukou (China), and Stuttgart (Germany) (Supplementary Fig. 4). The urban ratios (within the 1° × 1° domain) for the five cities are comparable, i.e., approximately 14%, but the spatial arrangements of urban elements (such as buildings and roads) vary significantly among these cities. The landscape shape index for the five urban footprints is 7.1, 8.9, 17.2, 17.9, and 22.0, respectively. Note that larger landscape shape index points to more spatially disaggregated urban elements.

Austin and San Antonio represent compact cities with a single prominent city center. There are no sub-centers or small towns surrounding the city's center (Supplementary Fig. 4a and Fig. 4b). The difference between Austin and San Antonio lies in city shape. Urban elements in Austin are arranged in a circular pattern around a central point (i.e., a Circular city), while these elements are spatially distributed along a corridor for San Antonio (i.e., a Ribbon city). The impact of city shape on rainfall patterns has been examined previously[8]. Both Fuyang and Stuttgart share a similar urban footprint characterized by a significant city center encircled by towns and smaller cities (and comparable landscape shape indexes). The distinction lies in the placement of the prominent city center; it's positioned at the heart of Fuyang (i.e., a Satellite city, Supplementary Fig. 4c), while it's situated on the outskirts of Stuttgart (an Edge city, Supplementary Fig. 4d). The two cities have similar landscape shape indices as well. For Zhoukou, the patches of urban elements are of comparable sizes, and are evenly distributed across the domain (i.e., a Scatter city, Supplementary Fig. 4e). The landscape shape index is the largest of all five urban footprints. Fuyang, Stuttgart, and Zhoukou represent the urban footprint of dispersed cities.

The five urban scenarios are chosen by following the suggested spatial models of city structure in sustainable development (ref. 41) as well as the existence of these prototypes in real-world cities (through manually examining the global urban cover map). These selections of city prototypes are not intended to be exhaustive, but rather to represent a subset of possible urban footprints. As the domain size increases, more types of urban footprints may emerge.

## Numerical simulations

We examine the impacts of different urban footprints on rainfall patterns through the Real Atmosphere, Ideal Land surface (RAIL) simulations. The RAIL simulations are based on the Weather Research and Forecasting (WRF) model (version 3.9.1, ref. 42). The atmospheric component of these simulations is initialized with three-dimension, heterogeneous variables, while the land component is kept intentionally simple. There are only two land use/land cover types: crop and urban (i.e., high-density residential) in the innermost model domain (referred to as model domain). The terrain is set completely flat (32 m above sea level) with spatially uniform soil properties (e.g., soil type, soil moisture, etc.). The soil type is set to silty clay loam. There is a "no-city" scenario with homogenous, crop-only land use in the model domain. Urban scenarios are then configured by replacing crop land use with different urban footprints (i.e., Circular, Ribbon, Satellite, Edge, and Scatter) within a central area of 101 × 101 grids (referred to as the urban domain) in the model domain. The cropland type is the same for all scenarios. We manually choose and remove small patches of urban grids, to ensure that the urban ratios are exactly the same (i.e., 14.1%) for the five urban scenarios. The only difference among them is how urban elements are horizontally organized. The single-layer Urban Canopy Model together with the Noah land surface model is used to represent exchanges of heat, moisture, and momentum between urban canopy and lower atmosphere. We set all urban grids as Industrial and Commercial land use. We adopt the default urban canopy model parameters (see Supplementary Table 3 for details).

We simulate an extreme rainfall event with strong synoptic forcings (mostly responsible for extreme rainfall in urban environments) over central China from 20-21 August 2012. Total rainfall accumulation exceeds 100 mm for 13 in-situ rain gauges, with maximum 1-h rainfall intensity of 115.8 mm. The storm environment is featured with the passage of a low vortex and a moist southwesterly jet along the shear line. The difference of equivalent potential temperature between 500 hPa and 850 hPa is −6.2 °C, indicating strong potential for convection. The maximum convective available potential energy is 751 J kg$^{-1}$.

Initial and boundary conditions for the simulations are represented by the NCEP Final Operational Global Analysis product, with a spatial resolution of 1 degree and a temporal resolution of 6 h. We configure three one-way nested domains. The horizontal grids are 200 × 200, 190 × 190, and 178 × 178, with grid spacing of 9 km, 3 km, and 1 km, respectively. The 1-km domain centers over 32.9 °N and 115.82 °E. The soil moisture ranges from 0.24 to 0.30 m$^3$ m$^{-3}$ (vary with depths) over the model domain at the beginning of the simulations. We configure 38 vertical levels in the model, with 20 of them below 2 km above the ground. We set 50 hPa as the upper boundary.

The WRF physics options include: The Rapid Radiative Transfer Model for long-wave radiation and Dudhia's scheme for short-wave radiation. The cumulus scheme is turned off for all domains due to the fine spatial resolution of horizontal grids (less than 10 km). We carry out ensemble simulations for both the "control" scenario and five urban scenarios by adopting combinations of three different microphysical schemes (WSM6, Thompson graupel, and Morrison double-moment scheme) and two different planetary boundary layer schemes (YSU and Mellor-Yamada-Janjic (Eta) TKE scheme), while maintaining the rest of the physics options. This results in six ensemble members for each land use scenario. The differences in the ensemble mean between five urban scenarios and the "no-city" scenario can shed light on the impacts of different urban footprints on rainfall.

All the simulations are initiated at 00 UTC 19 August 2012, and run for 72 h. The output interval is 1 h. Rainfall episode spans 24 h, from 00 UTC 20 till 00 UTC 21 August. The first 12 h of the simulations are regarded as the spin-up period. Pre-storm period is from 12 UTC 19 August to 00 UTC 20 August.

## Buffer analysis

We highlight distinct rainfall patterns across five urban scenarios based on buffer analysis (i.e., around the boundary of each individual urban coverage). The rainfall anomalies between each of the five urban scenarios and "no-city" scenario are first calculated. The anomalies are subsequently averaged over the urban grids and their surrounding regions, with the buffer distance gradually increasing, until the entire model domain is encompassed.

## Data availability

The GAIA product is available at http://data.starcloud.pcl.ac.cn/zh. The SRTM30+ Global 1-km Digital Elevation Model is available at https://catalog.data.gov/dataset/srtm30-global-1-km-digital-elevation-model-dem-version-11-land-surface. The list of global populated areas is available at https://www.naturalearthdata.com/downloads/10m-cultural-vectors/10m-populated-places/. The IMERG (Version 06) rainfall product is available at https://gpm1.gesdisc.eosdis.nasa.gov/data/GPM_L3/GPM_3IMERGDF.06/. The NCEP Final Operational Global Analysis product is available at https://rda.ucar.edu/datasets/ds083.2/. The RAIL simulation outputs are available from the corresponding author upon request.

## Code availability

The source code for this study is available from https://doi.org/10.6084/m9.figshare.24303136.v1. The Python packages GDAL and PyLandStats can be downloaded from https://gdal.org/api/python_bindings.html and https://pylandstats.readthedocs.io/en/latest/.

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

## Acknowledgements

L.Y., Y.Y., and Y.S. are supported by the Basic Research Program of Jiangsu Province (Grant No. BK20231541) and National Natural Science Foundation of China (Grant No. 52379012). The numerical simulations in this paper are implemented on the computing facilities in the High-Performance Computing Center (HPCC) of Nanjing University.

## Author contributions

L.Y. initiated the study. L.Y. designed the research and wrote the first draft of the paper. Y.Y., Y.S. and G.Z. conducted empirical analyses related to the global rainfall product and urban clusters. L.Y., J.Y., J.S., and D.N. contributed to the interpretation of the results, writing and editing of the manuscript.

## Competing interests

The authors declare no competing interests.
