## [Peer Review File · Nature Communications]

Urban development pattern's influence on extreme rainfall occurrencesREVIEWER COMMENTS

Reviewer #1 (Remarks to the Author):

This study endeavors to unveil the connection between urban development patterns and rainfall occurrences through an analysis encompassing over 1700 cities worldwide. The research posits that more compact cities tend to experience increased rainfall. While the paper is intriguing and well-crafted, my skepticism persists due to the following concerns regarding the reliability of the author's conclusions.

1. Could a spatial resolution of $1^\circ \times 1^\circ$ be considered somewhat limited for urban-scale studies? Taking into account that the radius of a city usually extends just a few tens of kilometers, it's worth noting that the entire city might be represented by only one or two pixels in your current study resolution. There appears to be a notable opportunity for the authors to explore urban scales with a higher resolution.

2. The authors should consider delving deeper into the quantification of urban development patterns. Currently, the study relies solely on the landscape shape index to measure urban aggregation. However, it is crucial to acknowledge that numerous other indicators exist for this purpose. It is recommended that the authors broaden their analysis by comparing the outcomes obtained from a variety of indicators. This approach will enhance the overall robustness of their conclusions.

3. The authors concentrate exclusively on the influence of the landscape shape index of cities on rainfall, overlooking the three-dimensional configuration of cities. In reality, variations exist in the distribution and altitude changes of cities across different regions. The height of cities serves as a significant indicator impacting surface roughness and convection. It is imperative for the authors not to overlook the three-dimensional morphology of cities when investigating the impacts of urbanization on rainfall.

4. The authors should incorporate a discussion regarding the specific climate of each city. For instance, cities situated in humid regions typically experience higher precipitation compared to those in arid areas, irrespective of their urban form. The authors' classification of cities solely based on urban ratios and urbanization rates, without accounting for additional factors such as climate type, may introduce bias into the results. It is essential to consider and control for other related factors to ensure a more comprehensive and accurate analysis.

5. I harbor reservations regarding the reliability of the authors' conclusions. While the authors assert that their data indicate higher rainfall in more agglomerated cities, the observed differences appear quite marginal (e.g., 1.5% vs. 0.5%). Given that the authors' analysis lacks effective control for various factors, as noted in our previous comments, my skepticism grows regarding the dependability of these observed distinctions. A more robust consideration and control of additional variables are crucial for instilling confidence in the validity of the reported findings.

Reviewer #2 (Remarks to the Author):

Review – Extreme rainfall occurrences linked more to compact than dispersed urban development

Using satellite-based rainfall and land cover data, the authors show that levels of spatial aggregation in urban developments, in addition to urban coverage, affect changes in rainfall extremes. They also perform short simulations with a convection-permitting model to help with the understanding of mechanisms that lead to the differences in rainfall using 5 urban footprints based on real cities, linking the rainfall differences to urban heat island intensity and low-level convergence. This is a very

interesting piece of research and the results are of key importance in the adaptation to climate change as it shows that urban planning needs to be taken into account when managing flood risk. The methodology is appropriate and is described sufficient detail, however, some refining of the manuscript is required in order to effectively communicate the findings. At times it is not clear which figure the text is referring to and there are some statements that are not necessarily linked to the analysis. The authors should be clear about which statements are conclusions from their analysis versus those which are simply a possible interpretation of the results. Please see detailed comments below for specific examples.

Line 24: is flood hazard included in this study? It is also mentioned in the introduction but only rainfall is considered in the analysis.

Line 27: be clear about the meaning of 'rainfall anomalies' in this context.

Lines 25 to 27: missing word after 'cities'. Please add 'that' or rephrase.

Lines 26, 30, 43, 46, 69, 70, 77, 82, 97, 122, 138, 225, 276: Words downtown, potential, understanding, terrain, coverage, occurrence, organization, vicinity, relief should be singular.

Line 54: Please change 'Contrast' to 'contrasting'

Line 66: cities are the most vulnerable compared with what?

Line 72: 'but tendencies' please rephrase.

Line 74 to 76: Sentence beginning "convection-permitting.." needs rephrasing.

Line 80: 'first-time'. No hyphen needed here.

Lines 107 to 109: The 'Methods' section states that increased LSI is associated with a more dispersed form of urbanisation, however, lines 107 to 109 in 'Results' suggests that decreased LSI is evidence for a more dispersed pattern. Please clarify.

Lines 133 to 134: Please refer to a figure here when mentioning differences in urbanization rates.

Lines 158 to 160: Which figure is being referred to here?

Lines 160 to 162: Has this been shown in the analysis? If so, which figure?

Line 163: does figure S5 show domain-average rainfall? It appears to show urban and rural values only.

Line 171: Where does the value of 2 hours come from?

Lines 176 to 178: does 'compact city scenarios' mean circular and ribbon? If so, please specify. Please also refer to Figure 2f if that is the relevant figure.

Lines 178 to 179: Fig S5 shows the differences in total rainfall so I cannot see how the statement about frequent extreme rainfall is supported by this figure.

Line 183: a reference to Figure 2f would be useful here.

Line 192: Please change to 'Contrasting urban signatures...'

Lines 194 to 196: text is inconsistent with the Fig S11.

Line 205: Is there any plot that shows the vertical velocity at 850hPa?

Lines 210 to 212: which figure(s) are relevant here?

Line 219: Please change to 'Contrasting impacts.....patterns...'

Line 244: replace prioritized with 'important'?

Line 256: words missing after 'our analysis'

Line 296: "best practices of existing global climate models". Please rephrase.

Lines 330 to 341: Here it would be useful to refer to a figure or figures when describing the urban footprints.

Line 344: 'footprints' should be singular here.

Line 367 to 368: please expand a little on the last sentence.

Line 393: replace 'till' with until

Figure 2: What are the units of the colour bar? Is this the change ratio between the two time periods?

Figure 3 caption: It would be clearer to refer to the points as 'circles' as opposed to scatter.

Figure S3 caption: is there a word missing after contiguous? Also singular of indices is 'index'.

Figure S4: the colour map makes it difficult to see the urban footprints against the total rainfall. Please could the colour map be changed so that the predominant colour is lighter?

Figure S7: the captions says that panels a and b are for the innermost domain but plots are labelled

with 'outer domain'.

Figure S9: as for Fig S7, the labelling of plots contradicts the caption. Also there is a typo in the title of plot (b).

Figure S11: typo in title of panel a. x-labels are not consistent with scenario names in Fig S10.

Reviewer #3 (Remarks to the Author):

Review: Major revision

L23: the environment implication ... this is too general please specify related to the current study

L24: what the reason for this large number of cities? Any physical argument that can be used here?

L31: The last sentence "Global climate models...change impact" seems disconnected from the rest of the study and also not giving any new outcome as it is already known that GCM cannot represent urban-atmosphere interaction correctly.

L31: no physical reason is put in the abstract for the reason why compact influence the extreme precipitation differently from dispersed development.

L48: should indicate also the dependency to the background climate and to the aerosols which impact a lot the precipitation in urban environment.

L52: this reference is not related to the previous discussion as this reference is for the interaction of UHI and heat waves

L65: add the newest reference from UCCRN ARC3.2: Bader, D. A., Blake, R., Grimm, A., Hamdi, R., Kim, Y., Horton, R., and Rosenzweig, C. (2018). Urban climate science. In Rosenzweig, C., W. Solecki, P. Romero-Lankao, S. Mehrotra, S. Dhakal, and S. Ali Ibrahim (eds.), *Climate Change and Cities: Second Assessment Report of the Urban Climate Change Research Network*. Cambridge University Press, New York, 27-60.

L90: the result for GCM looks disconnected from the rest of the text and as said before is not new finding.

L96: Figure 1 is very strange that no value are plotted for south east China, south Korea and Japan while increase in urban areas has been observed by global dataset: Kuang W, Du G, Lu D, Dou Y, Li X, Zhang S, Chi W, Dong J, Chen G, Yin Z, Pan T, Hamdi R, Hou Y, Chen C, Li H, Miao C. Global observation of urban expansion and land-cover dynamics using satellite big-data. *Sci Bull (Beijing)*. 2021 Feb 26;66(4):297-300. doi: 10.1016/j.scib.2020.10.022. Epub 2020 Nov 3. PMID: 36654403.

L112: What is the impact of having different numbers for the 3 categories: I 1105, II 417, and III 268. Is the result the same if for example you use only a part of group I in order to have similar number to groups II and III.

L125: any explanation why in the three cases the precipitation anomalies are located south east of the urban areas?

L146: is the result depending on the background climate of the selected cities, I would suggest to check the impact of the background climate on the results for the 3 groups of cities another factor could be also studied is the aerosols effect. This will allow to check if the results are not dependent of the background climate of the urban areas. Maybe the aerosols effect can be a bit difficult to take into account but the interplay between the background climate and the urban development on the extreme precipitation over urban areas can be of a great interest to the community.

L151: is the crop land the same for all cities or depending on the climate of the cities? Can this affect your results?

L153: are those 5 scenarios representing of the full set of all 1790 cities? How it has been chosen to use these 5 scenarios?

L158: we cannot see the difference through the cot domain is this related to FigS4?

L167: Fig S5: any explanation why the impact is large for the Ribbon scenario? Is this related to the size of the simulation domain with respect to the spatial distribution of the urban grid?

L202: FigS11 should be corrected, dispersed should be replaced by scattered and title corrected

“Urabn”

L230: the discussion on GCM seems disconnected from the rest of the manuscript and no result support the already known statement.

L255: what is the sensitivity of your results to changing this threshold of 20% for the urban grid?

L315: how they are resampled, we know that this treatment can affect the extreme values.

L324: how the cities are selected, it might be good to select cities from different background climate, it will give more weight to the result.

L358: please add information on the urban types used for the single-layer urban canopy with the parameter for the building used such as height of the buildings, heat capacity ...etc

L384: it might be very good to add also uncertainty related to soil initialization for both urban and rural areas since it can affect the simulation and then widening the uncertainties of the ensemble.

Response to Reviewer #1

(1) This study endeavors to unveil the connection between urban development patterns and rainfall occurrences through an analysis encompassing over 1700 cities worldwide. The research posits that more compact cities tend to experience increased rainfall. While the paper is intriguing and well-crafted, my skepticism persists due to the following concerns regarding the reliability of the author's conclusions.

Response: Thank you for the appreciation of our manuscript. We further test the sensitivity of our conclusions to a couple of factors (i.e., landscape shape index, background climate) by following your suggestions and address our responses below in details. Thanks!

(2) Could a spatial resolution of $1^\circ \times 1^\circ$ be considered somewhat limited for urban-scale studies? Taking into account that the radius of a city usually extends just a few tens of kilometers, it's worth noting that the entire city might be represented by only one or two pixels in your current study resolution. There appears to be a notable opportunity for the authors to explore urban scales with a higher resolution.

Response: The resolution of our analyses is largely constrained by the availability of high-resolution, global-scale rainfall products. The grid spacing of IMERG, i.e., 0.1° , is the best available. Continental or regional scale analyses based on rainfall products with higher spatial resolutions, such as the Multi-Radar/Multi-Sensor precipitation reanalysis over United States (~ 1 km), can be pursued in future. We made this clear in the revised manuscript. Thanks!

(3) The authors should consider delving deeper into the quantification of urban development patterns. Currently, the study relies solely on the landscape shape index to measure urban aggregation. However, it is crucial to acknowledge that numerous other indicators exist for this purpose. It is recommended that the authors broaden their analysis by comparing the outcomes obtained from a variety of indicators. This approach will enhance the overall robustness of their conclusions.

Response: We agree with you that there are numerous indicators to measure urban aggregation. Each of them demonstrates different aspects in terms of urban aggregation

pattern. It is impossible to test them in an exhaustive way. To test the robustness of our conclusion, we choose another **three dimensionless indicators**, i.e., number of patches, patch densities, and splitting index. We show that these indicators are significantly correlated with landscape shape index.

Figure R1a. Scatterplots between three another aggregation indices and landscape shape index.

The clustering results show three different city groups (Figure R1b, e.g., based on number of urban patches). The contrasting rainfall patterns persist among the three city groups based on the new aggregation index (Figure R1c).

Figure R1b. Differences in mean urban ratio and number of patches (as the aggregation index) across three city groups. Our clustering analysis show three as the optimal number of clusters.

Figure R1c. Composite mean change ratios in extreme rainfall (i.e., exceeding the 90th percentile daily rainfall of rainy days) frequencies for different city groups with diverse development patterns between the period 2000-2005 and 2016-2020 (results based on number of patches as the aggregation index).

We prefer not to include all the results into supplementary files, since there are already 17 figures and 3 tables. However, we do mention the consistency of our results using different aggregation indices in the revised manuscript. Thanks!

(4) The authors concentrate exclusively on the influence of the landscape shape index of cities on rainfall, overlooking the three-dimensional configuration of cities. In reality, variations exist in the distribution and altitude changes of cities across different regions. The height of cities serves as a significant indicator impacting surface roughness and convection. It is imperative for the authors not to overlook the three-dimensional morphology of cities when investigating the impacts of urbanization on rainfall.

Response: We realize that the development of products that accurately capture and characterize the three-dimensional morphology of global cities and their dynamics is still in its early stages. For instance, we do not have building heights in early 2000s, and thus cannot quantify the changing urban development pattern in vertical dimension. However, we believe it is fair to expect that changes in building heights positively correlate with changes in building densities in the real world, i.e., increases in urban density are accompanied by increase in building heights. Our two-dimensional characterization thus maybe sufficient to demonstrate contrasting urban development patterns across global cities. Even so, we mention this as a caveat in our revised manuscript and suggest a future endeavor. Thanks!

Lines 249-251: “...For instance, the emerging satellite-based products of global building footprint products can provide valuable three-dimensional characterization of global

cities and their temporal evolution patterns (Marconcini et al., 2021)”

Lines 308-312: “...We note that only two-dimensional characteristics of urban development patterns are considered here. This is because development of products that capture and characterize the three-dimensional morphology of global cities and their dynamics is in its early stages. For instance, characterizing three-dimension urban morphology of global cities during early 2000s is still not feasible. Three-dimensional characterization of urban development is worthwhile to be examined in future.”

(5) The authors should incorporate a discussion regarding the specific climate of each city. For instance, cities situated in humid regions typically experience higher precipitation compared to those in arid areas, irrespective of their urban form. The authors' classification of cities solely based on urban ratios and urbanization rates, without accounting for additional factors such as climate type, may introduce bias into the results. It is essential to consider and control for other related factors to ensure a more comprehensive and accurate analysis.

Response: This is a good suggestion! We use four climate types (i.e., tropical, dry, temperate, continental) corresponding to the first level of the Köppen climate classification. There are no cities in polar climate. We determine the background climate type based on the location of each city. We find that the contrasting rainfall patterns across different city groups persist in different climate types (see Figure R2 below). We made this clear in the revised manuscript. Thanks!

Figure R2. Contrasting rainfall patterns across different city groups and background climates. The number shown in each subplot indicate number of cities belonging to each category.

(6) I harbor reservations regarding the reliability of the authors' conclusions. While the authors assert that their data indicate higher rainfall in more agglomerated cities, the observed differences appear quite marginal (e.g., 1.5% vs. 0.5%). Given that the authors' analysis lacks effective control for various factors, as noted in our previous comments, my skepticism grows regarding the dependability of these observed distinctions. A more

robust consideration and control of additional variables are crucial for instilling confidence in the validity of the reported findings.

Response: Thank you for this critique. Note that here the ratio is defined as $(R2-R1)/(R2+R1)$, where R1 and R2 represents rainfall statistic in 2001-2005 and 2016-2020 period, respectively. This is different from the definition of relative change, i.e., $(R2-R1)/R1$, which should lead to larger changes. The ratio can be inflated by a large sample of cities that experienced weak urbanization rates during the 2001-2020 period. According to the Clausius-Clapeyron scaling relation, we expect a 7% increase in rainfall rate per 1 degree increase in air temperature. The increase in global mean temperature is approximately 0.5 degree (refer to https://www.ncei.noaa.gov/access/monitoring/climate-at-a-glance/global/time-series/globe/land_ocean/12/12/2000-2020), we thus expect 3.5% increase in rainfall rate. Urban impact on rainfall is of a comparable magnitude with that associated with increase in global temperature.

To enhance the robustness of our results, we carry out sensitivity analyses that focus on landscape shape index, background climate by following your previous comments and some comments by the other two reviewers. We also test the sensitivity of our results by varying the thresholds to define an urban grid (see Response to Comment #19 by Reviewer #3). Thanks!

Response to Reviewer #2

(1) Using satellite-based rainfall and land cover data, the authors show that levels of spatial aggregation in urban developments, in addition to urban coverage, affect changes in rainfall extremes. They also perform short simulations with a convection-permitting model to help with the understanding of mechanisms that lead to the differences in rainfall using 5 urban footprints based on real cities, linking the rainfall differences to urban heat island intensity and low-level convergence. This is a very interesting piece of research, and the results are of key importance in the adaptation to climate change as it shows that urban planning needs to be taken into account when managing flood risk.

Response: Thank you for the appreciation of our manuscript!

(2) The methodology is appropriate and is described sufficient detail, however, some refining of the manuscript is required in order to effectively communicate the findings. At times it is not clear which figure the text is referring to and there are some statements that are not necessarily linked to the analysis. The authors should be clear about which statements are conclusions from their analysis versus those which are simply a possible interpretation of the results. Please see detailed comments below for specific examples.

Response: Thank you for this general critique. We revise the manuscript by following your comments and address them below on a point-to-point basis. We especially go through the manuscript to make sure each statement is linked to our analysis, and each figure is properly referred to in the text. Thanks!

(3) Line 24: is flood hazard included in this study? It is also mentioned in the introduction but only rainfall is considered in the analysis.

Response: We agree with you that mentioning flood hazard seems 'out-of-context', since as the title suggests that extreme rainfall is the only focus. We thus remove it from the abstract section. We, however, prefer to maintain relevant texts in the Introduction section to highlight the implication of our analysis to flood risk management in cities. Thanks!

(4) Line 27: be clear about the meaning of 'rainfall anomalies' in this context.

Response: We mean "anomalies in extreme rainfall frequency". Thanks!

(5) Lines 25 to 27: missing word after 'cities'. Please add 'that' or rephrase.

Response: we add 'that' after 'cities'.

(6) Lines 26, 30, 43, 46, 69, 70, 77, 82, 97, 122, 138, 225, 276: Words downtown, potential, understanding, terrain, coverage, occurrence, organization, vicinity, relief should be singular.

Response: All revised as suggested.

(7) Line 54: Please change 'Contrast' to 'contrasting'

Response: Revised as suggested.

(8) Line 66: cities are the most vulnerable compared with what?

Response: We rephrase the sentence. It now reads "*...cities are particularly vulnerable to extreme rainfall and flooding under a changing climate...*". Thanks!

(9) Line 72: 'but tendencies' please rephrase.

Response: The sentence now reads "*...but this expansion tends to be characterized by dispersed urban development.*" Thanks!

(10) Line 74 to 76: Sentence beginning "convection-permitting.." needs rephrasing.

Response: The sentence now reads "*Convection-permitting modeling analysis shows that spatially aggregated urban footprints (i.e., compact cities) pose strong thermodynamic and aerodynamic disturbances to synoptic forcings.*"

(11) Line 80: 'first-time'. No hyphen needed here.

Response: Revised as suggested.

(12) Lines 107 to 109: The 'Methods' section states that increased LSI is associated with a more dispersed form of urbanisation, however, lines 107 to 109 in 'Results' suggests that decreased LSI is evidence for a more dispersed pattern. Please clarify.

Response: Thank you for catching this typo! It should be "increased".

(13) Lines 133 to 134: Please refer to a figure here when mentioning differences in urbanization rates.

Response: We refer to Fig. S1. Thanks!

(14) Lines 158 to 160: Which figure is being referred to here?

Response: Fig. S5 should be referred to here. We made this clear in the manuscript. Thanks!

(15) Lines 160 to 162: Has this been shown in the analysis? If so, which figure?

Response: You are correct. This sentence is out of context. We removed it in the revised manuscript. Thanks!

(16) Line 163: does figure S5 show domain-average rainfall? It appears to show urban and rural values only.

Response: We apologize for referring to an incorrect figure. We replace it with a correct one in the revised manuscript. Thanks!

(17) Line 171: Where does the value of 2 hours come from?

Response: We check the values in Figure 2 (differences among WRF scenarios). It can also be interpreted from the color bar. We do not modify the text. Thanks!

(18) Lines 176 to 178: does 'compact city scenarios' mean circular and ribbon? If so, please specify. Please also refer to Figure 2f if that is the relevant figure.

Response: Done as suggested. Thanks!

(19) Lines 178 to 179: Fig S5 shows the differences in total rainfall so I cannot see how the statement about frequent extreme rainfall is supported by this figure.

Response: extreme rainfall frequency is actually shown in Figure 2. Our rationale is, total rainfall is dominated by extreme rainfall frequency multiplied by extreme rain rate (2 mm/h or 10 mm/h). The increased total rainfall over urban grids as seen in Fig. S7 is thus associated with increases in extreme rainfall frequency. We modify the text accordingly . Thanks!

(20) Line 183: a reference to Figure 2f would be useful here.

Response: Done as suggested. Thanks!

(21) Line 192: Please change to ‘Contrasting urban signatures...’

Response: Done as suggested. Thanks!

(22) Lines 194 to 196: text is inconsistent with the Fig S11.

Response: We replace Fig. S11 with a correct figure. Thanks for pointing this out.

(23) Line 205: Is there any plot that shows the vertical velocity at 850hPa?

Response: We only show frequency of positive velocity at 850 hPa in Fig. S13. We additionally show a scatter plot that indicate the relationship between mean frictional velocity and velocity at 850 hPa in a Fig. S14 in the revised manuscript, as response to your request. Thanks!

(24) Lines 210 to 212: which figure(s) are relevant here?

Response: We refer to Fig. S12 in the revised manuscript. Thanks!

(25) Line 219: Please change to ‘Contrasting impacts.....patterns...’

Response: Done as suggested. Thanks!

(26) Line 244: replace prioritized with ‘important’?

Response: Done as suggested. Thanks!

(27) Line 256: words missing after 'our analysis'

Response: The sentence now reads "*This makes our analyses less contaminated*".

(28) Line 296: "best practices of existing global climate models". Please rephrase.

Response: We change "practices" to "practice" in the revised manuscript.

(29) Lines 330 to 341: Here it would be useful to refer to a figure or figures when describing the urban footprints.

Response: Thanks for this suggestion. We refer to Fig. S4 and its sub figures.

(30) Line 344: 'footprints' should be singular here.

Response: Done as suggested. Thanks!

(31) Line 367 to 368: please expand a little on the last sentence.

Response: We realize that the last sentence is out of context and remove it. Thanks!

(32) Line 393: replace 'till' with until

Response: Done as suggested. Thanks!

(33) Figure 2: What are the units of the color bar? Is this the change ratio between the two time periods?

Response: The unit of the color bar is in number of hours (with rain rate exceeding 10 mm/h). The change ratio is calculated similarly using equation (3) but for urban scenarios and no-city scenario. We made this clear in the caption. Thanks!

(34) Figure 3 caption: It would be clearer to refer to the points as 'circles' as opposed to scatter.

Response: Done as suggested. Thanks!

(35) Figure S3 caption: is there a word missing after contiguous? Also singular of indices is 'index'.

Response: Revised as suggested. Thanks!

(36) Figure S4: the colour map makes it difficult to see the urban footprints against the total rainfall. Please could the colour map be changed so that the predominant colour is lighter?

Response: Revised as suggested. Thanks!

(37) Figure S7: the captions says that panels a and b are for the innermost domain but plots are labelled with 'outer domain'.

Response: We apologize for this inconsistency. We revised the labels in the plots.
Thanks!

(38) Figure S9: as for Fig S7, the labelling of plots contradicts the caption. Also there is a typo in the title of plot (b).

Response: We apologize for this inconsistency. We revised the labels in the plots.
Thanks!

(39) Figure S11: typo in title of panel a. x-labels are not consistent with scenario names in Fig S10.

Response: Revised as suggested. Thanks!

Response to Reviewer #3

(1) L23: the environment implication ... this is too general please specify related to the current study.

(2) L24: what the reason for this large number of cities? Any physical argument that can be used here?

(3) L31: The last sentence “Global climate models...change impact” seems disconnected from the rest of the study and also not giving any new outcome as it is already known that GCM cannot represent urban-atmosphere interaction correctly.

(4) L31: no physical reason is put in the abstract for the reason why compact influence the extreme precipitation differently from dispersed development.

Response: Thank you for these detailed suggestions! We reconstruct the abstract in our revised manuscript. The new abstract is attached below for your reference.

Abstract: Growing urban population and the distinct strategies to accommodate them lead to diverse urban development patterns worldwide. While local evidence suggests the presence of urban signatures in rainfall anomalies, there is limited understanding of how rainfall responds to divergent urban development patterns worldwide. Here we unveil a divergence in the exposure to extreme rainfall for 1,790 inland cities globally, attributable to their respective urban development patterns. Cities that experience compact development tend to witness larger increases in extreme rainfall frequency over downtown than their rural surroundings, while the anomalies in extreme rainfall frequency diminish for cities with dispersed development. Convection-permitting simulations further suggest compact urban footprints lead to more pronounced urban-rural thermal contrasts and aerodynamic disturbances. This is directly responsible for the divergent rainfall responses to urban development patterns. Our analyses offer significant insights pertaining to the priorities and potential of city-level efforts to mitigate the emerging climate-related hazards, particularly for developing countries experiencing rapid urbanization.

(5) L48: should indicate also the dependency to the background climate and to the aerosols which impact a lot the precipitation in urban environment.

Response: Good point! We add three references(Lalonde, Oudin, and Bastin 2023; Sarangi et al. 2018; Vo et al. 2023) to support this argument. Thanks!

Lalonde, Morgane, Ludovic Oudin, and Sophie Bastin. 2023. "Urban Effects on Precipitation: Do the Diversity of Research Strategies and Urban Characteristics Preclude General Conclusions?" *Urban Climate* 51(May 2022):101605.

Sarangi, Chandan, S. N. Tripathi, Yun Qian, Shailendra Kumar, and L. Ruby Leung. 2018. "Aerosol and Urban Land Use Effect on Rainfall Around Cities in Indo-Gangetic Basin From Observations and Cloud Resolving Model Simulations." *Journal of Geophysical Research: Atmospheres* 123(7):3645–67.

Vo, Thuy Trang, Leiqiu Hu, Lulin Xue, Qi Li, and Sisi Chen. 2023. "Urban Effects on Local Cloud Patterns." *Proceedings of the National Academy of Sciences* 120(21):1–11.

(6) L52: this reference is not related to the previous discussion as this reference is for the interaction of UHI and heat waves

Response: Agreed. We delete the sentence in the revised manuscript.

(7) L65: add the newest reference from UCCRN ARC3.2: Bader, D. A., Blake, R., Grimm, A., Hamdi, R., Kim, Y., Horton, R., and Rosenzweig, C. (2018). Urban climate science. In Rosenzweig, C., W. Solecki, P. Romero-Lankao, S. Mehrotra, S. Dhakal, and S. Ali Ibrahim (eds.), *Climate Change and Cities: Second Assessment Report of the Urban Climate Change Research Network*. Cambridge University Press, New York, 27-60.

Response: Done as suggested. Thanks!

(8) L90: the result for GCM looks disconnected from the rest of the text and as said before is not new finding.

Response: Agreed. We delete the sentence in the revised manuscript.

(9) L96: Figure 1 is very strange that no value are plotted for south east China, south Korea and Japan while increase in urban areas has been observed by global dataset: Kuang W, Du G, Lu D, Dou Y, Li X, Zhang S, Chi W, Dong J, Chen G, Yin Z, Pan T, Hamdi R,

Hou Y, Chen C, Li H, Miao C. Global observation of urban expansion and land-cover dynamics using satellite big-data. *Sci Bull (Beijing)*. 2021 Feb 26;66(4):297-300. doi: 10.1016/j.scib.2020.10.022. Epub 2020 Nov 3. PMID: 36654403.

Response: Thank you for this critique. We checked the robustness of our results. Cities in southeastern China are not included due to the proximity of complex terrain (elevation difference exceeds 500 m). Cities in south Korea and Japan are not included due to the proximity of land-water boundary and/or complex terrain. Our future studies will include complex urban environments, but only focus simple inland cities in the present study. In addition, we do not include cities with no changes in urban coverages during 2003-2018. Thanks!

(10) L112: What is the impact of having different numbers for the 3 categories: I 1105, II 417, and III 268. Is the result the same if for example you use only a part of group I in order to have similar number to groups II and III.

Response: Thank you for this critique! We test the sensitivity of the spatial rainfall pattern of Group 1 on group size. We found little variations to the spatial rainfall pattern. We add details of our sensitivity experiments and a new figure (Fig. S17, shown below) in the revised manuscript.

Supplementary Fig. 1. Profiles of the changes in extreme rainfall frequency for city Group I by randomly selecting (a, b) 200, and (c, d) 400 cities from the group. The selection is done 100 times. Grey shade represents the range between 25 and 75 percentiles of the 100 composite mean profiles. Black line shows the composite mean profile for the entire group (see Figure 1b for the spatial pattern).

(11) L125: any explanation why in the three cases the precipitation anomalies are located south east of the urban areas?

Response: According to the monthly wind climatology (see links below), the prevalent low-level wind during extreme rainfall period (June-July for Northern hemisphere and November-December for Southern hemisphere) is in west or northwest direction (with only few exceptions of cities located near the Equator). Previous studies agree with the urban rainfall enhancement in downwind region (e.g., Liu and Niyogi, 2019), we thus expect rainfall anomalies for the three cases in south east of urban areas. Thanks!

Monthly wind climatology during June:

https://iridl.ldeo.columbia.edu/maproom/Global/Climatologies/Vector_Winds.html?P=925&T=Jun

Monthly wind climatology during December:

https://iridl.ldeo.columbia.edu/maproom/Global/Climatologies/Vector_Winds.html?P=925&T=Dec

(12) L146: is the result depending on the background climate of the selected cities, I would suggest to check the impact of the background climate on the results for the 3 groups of cities another factor could be also studied is the aerosols effect. This will allow to check if the results are not dependent of the background climate of the urban areas. Maybe the aerosols effect can be a bit difficult to take into account but the interplay between the background climate and the urban development on the extreme precipitation over urban areas can be of a great interest to the community.

Response: We use four climate types (i.e., tropical, dry, temperate, continental, polar) corresponding to the first level of the Köppen climate classification. There are no cities in polar climate. We determine the background climate type based on the location of each city. We find that the contrasting rainfall patterns across different city groups persist in different climate types (see Figure R3 below). We made this clear in the revised manuscript. Thanks!

Figure R3. Contrasting rainfall patterns across different city groups and background climates. The number shown in each subplot indicate number of cities belonging to each category.

(13) L151: is the crop land the same for all cities or depending on the climate of the cities? Can this affect your results?

Response: Correct. The cropland (a land use type in the WRF model) is the same for all scenarios. We made this clear in the revised manuscript.

(14) L153: are those 5 scenarios representing of the full set of all 1790 cities? How it has been chosen to use these 5 scenarios?

Response: It is most likely impossible to have those 5 scenarios represent the full collection of ~1700 cities. For instance, the city shape can vary significantly in the compact city scenario. We chose the five scenarios by following the suggested spatial models of city structure in sustainable development (Reddy and Ulgiati 2015) as well as the existence of these prototypes in real-world cities (through manually examining the global urban cover map). Our idea is to show contrasting rainfall patterns between compact and dispersed urban patterns rather than represent the entire spectrum of urban scenarios. We made this clear in the revised manuscript. Thanks!

Reddy, B. Sudhakara and Sergio Ulgiati. 2015. "Energy Security and Development: The Global Context and Indian Perspectives." *Energy Security and Development: The Global Context and Indian Perspectives* (April 2015):1–519.

(15) L158: we cannot see the difference through the cot domain is this related to FigS4?

Response: The text should be referred to Fig. S5. We revised the text. Thanks!

(16) L167: Fig S5: any explanation why the impact is large for the Ribbon scenario? Is this related to the size of the simulation domain with respect to the spatial distribution of the urban grid?

Response: Both Circular and Ribbon are compact city scenarios. The differences between the two scenarios are associated with contrasting city shapes and their interactions with synoptic forcing. This aspect has been explored in our previous studies (Zhang et al. 2022). Larger impact for Ribbon is due to more notable disturbance on synoptic flows than Circular (as can be reflected low-level convergence in Figure 3b).

Zhang, Wufan, Jiachuan Yang, Long Yang, and Dev Niyogi. 2022. "Impacts of City Shape on Rainfall in Inland and Coastal Environments." *Earth's Future* 10(5):1–13.

(17) L202: FigS11 should be corrected, dispersed should be replaced by scattered and title corrected "Urabn"

Response: Revised as suggested. Thanks!

(18) L230: the discussion on GCM seems disconnected from the rest of the manuscript

and no result support the already known statement.

Response: Agreed. Our results do not address or support the inadequacy of global climate models. We delete this sentence. We, however, maintain the discussion on future directions in improving regional climate model simulations for urban climate studies. Thanks!

(19) L255: what is the sensitivity of your results to changing this threshold of 20% for the urban grid?

Response: We test the sensitivity of our cluster results by using different thresholds to define an urban grid, i.e., 10% and 30%. We find strong correlations between changes in urban coverages and LSI using the new thresholds and our original analysis (see Figure R4).

Figure R4. Scatterplots between changes in urban coverage and LSI using different thresholds (i.e., 0.1, and 0.3 versus 0.2) to define an urban grid. We show correlation coefficients in the subplots.

The consistency leads to little variations to our cluster results. Using different thresholds show three as the optimal choice of the cluster number, but larger (smaller) threshold leads to more (less) cities included in the clustering analyses. We show the

spatial pattern of three clusters and their corresponding rainfall pattern using the 10% (Figure R5) and 30% (Figure R6) as the threshold below.

Figure R5. Contrasting rainfall patterns across three city groups by using 10% to define an urban grid.

Figure R6. Contrasting rainfall patterns across three city groups by using 30% to define an urban grid.

We prefer not to include the results of our sensitivity analysis in the revised manuscript, as the supplementary materials seem a little bit overwhelming. In addition, 20% is frequently adopted to define an urban grid in existing land use/land cover datasets (e.g., the National Land Cover Dataset). Thanks!

(20) L315: how they are resampled, we know that this treatment can affect the extreme values.

Response: This is a fair point! To make our analysis less influenced by resampling uncertainty, we aggregate the urban footprint (in 30-m resolution) into the grid spacing of rainfall product (in 0.01-degree resolution), rather than the other way around. We made this clear in the revised manuscript. Thanks!

(21) L324: how the cities are selected, it might be good to select cities from different background climate, it will give more weight to the result.

Response: We chose the five scenarios by following the suggested spatial models of city structure in sustainable development as well as the existence of these prototypes in real-world cities (through manually examining the global urban cover map) (See our response to your Comment #14).

While these cities are indeed different in background climate (according to Köppen climate classification), we respectfully disagree that the selection of cities for numerical simulations should consider background climate. We designed and conducted single factor-controlled experiments that are only different in urban patterns.

Our empirical analyses (by following your suggestion, see Comment #12) additionally show little variations to the contrasting rainfall patterns on background climate. Thanks!

(22) L358: please add information on the urban types used for the single-layer urban canopy with the parameter for the building used such as height of the buildings, heat capacity ...etc.

Response: The urban category is Industrial and Commercial. We adopt the default urban canopy parameters. The values are summarized in Table S2. Thanks!

(23) L384: it might be very good to add also uncertainty related to soil initialization for both urban and rural areas since it can affect the simulation and then widening the uncertainties of the ensemble.

Response: We thank you for this suggestion. the soil type and all the associated soil properties are uniform in space across all scenarios. For instance, soil moisture ranges from 0.24 to 0.30 m³/m³ for all scenarios at the beginning of the simulations. Urban-rural contrasts are thus only tied to land use types and how urban elements are spatially organized. We made this clear in the revised manuscript. Thanks!

REVIEWERS' COMMENTS

Reviewer #2 (Remarks to the Author):

I am satisfied with the authors' response to my comments, although some more minor points have emerged from the revised manuscript.

Line 189 - I am finding the term 'innermost domain' difficult as the smallest domain seems to be referred to as the 'urban' domain. This is a little confusing.

Figure 3 - the caption does not mention delta T or convergence although these are the quantities on the axis labels. Also in the main text the authors refer to UHI and changes in convergence when talking about this figure (lines 210 to 211). Could the quantities be consistent between plot, caption and text?

Fig S13/S14 there appears to be inconsistent use of scatter/disperse as a categories.

Reviewer #3 (Remarks to the Author):

i am happy with the new version and the reply to my comments

Reviewer #4 (Remarks to the Author):

I have joined the reviewing process following the withdrawal of Reviewer 1, and my comments are not intended to conduct a complete review of the paper but rather to assess to what extent the concerns raised by the initial Reviewer 1 have been appropriately addressed by the authors in their revision.

Reviewer 1 expressed reservations about the suitability of a $1^{\circ} \times 1^{\circ}$ resolution for the proposed analysis. I acknowledge the data availability constraints outlined by the authors and the specific objectives and limitations of the analysis presented. However, I believe that the authors need to more thoroughly discuss the implications of these limitations in the "Implications" section of the paper, where the main research findings are presented. The authors should provide deeper reflections on the constraints of the analysis and propose potential strategies to address them. At the moment, the "Implications" section is quite simple and does not provide enough discussion and context, similar to the degree of complexity and analyses provided by the paper. This section, as the most relevant section of the paper, should be more solid.

Throughout the paper, the authors discuss the impact of urban form and development patterns on rainfall events, often referring to the relationship between cities and rainfall behavior. However, given the scale of the analysis ($1^{\circ} \times 1^{\circ}$), the findings are more applicable to large metropolitan areas and urban regions rather than individual cities. Therefore, I suggest that the authors revise the language throughout the paper to clearly indicate that they are working at an abstract scale, analyzing large urban regions and their aggregation patterns. The discussion should focus on debates about the agglomeration patterns of large urban regions rather than general debates about the shape and form of cities. Including references to specific literature on the analysis of the shape and aggregation of large urban regions would enhance the paper's discussion (in opposition to debates about compact / sprawl city models, which refer to the analyses of urban shapes at detailed scales, which are not done in this case).

I think that the landscape shape index is enough to analyse the urban aggregation at the chosen scale. In this regard, authors are right when saying that, although many other metrics exist, they usually provide similar information and are very correlated. However, this is caused by the coarser

scale of analysis, that hampers the characterization and measurement of urban aggregation pattern in a more accurate and precise way. Authors should acknowledge these limitations when discussing the results of the paper. As I explained above, it should be made clear, since the beginning, the objective and scale of the analysis that is carried out and its associated limitations (e.g. measuring urban aggregation at that scale can only be done through simple metrics such as the landscape shape index).

In a similar vein, although the analysis does not directly account for the urban height, this is not very relevant because of the objectives and scale of the analysis that is done. Although future studies taking into account this factor, once global data datasets providing this information become available, would be interesting, authors should make clear that their objective and analysis is very limited and just provides a specific general understanding of rainfall interaction with the pattern of urban aggregation, an understanding and conclusions that could be further enhanced through complementary and more detailed and ambitious studies.

A similar comment can be made regarding the contrasting climates of cities and their effects on the analyses. The authors provide a supplementary analysis that proves how this factor does not directly affect the obtained results. Again, they should acknowledge that part of this conclusion comes from the objective and scales of analysis, comparing a vast number of large urban regions at $1^{\circ} \times 1^{\circ}$ resolution, so only very general conclusions can be obtained.

All in all, the authors have effectively addressed the reviewer's comments. However, there remains a need for a more explicit acknowledgment of the main limitations identified by the reviewer within the paper and its discussion. The emphasis should not be on making extensive changes to the analyses conducted but rather on accurately framing the research question, explicitly outlining the limitations, and providing readers with a clear understanding of the study's contributions and the aspects that fall outside the scope of the analysis. This additional clarification will enhance the transparency and overall impact of the paper.

Response to Reviewer #2

(1) I am satisfied with the authors' response to my comments, although some more minor points have emerged from the revised manuscript.

Response: Thank you for the appreciation of our revised manuscript. We further improve the manuscript by addressing all the remaining points. Thanks!

(2) Line 189 - I am finding the term 'innermost domain' difficult as the smallest domain seems to be referred to as the 'urban' domain. This is a little confusing.

Response: You are right. We now refer innermost domain as 'model domain' in the revised manuscript to avoid misunderstandings. Thanks!

(3) Figure 3 - the caption does not mention delta T or convergence although these are the quantities on the axis labels. Also in the main text the authors refer to UHI and changes in convergence when talking about this figure (lines 210 to 211). Could the quantities be consistent between plot, caption and text?

Response: Thank you for this suggestion. We explain delta T and delta convergence in the caption of Figure 3. Delta T (i.e., temperature difference between urban scenario and no-city scenario) has a different meaning from UHI (i.e., urban-rural thermal contrast). We do not revise the text as suggested. Thanks all the same!

(4) Fig S13/S14 there appears to be inconsistent use of scatter/disperse as a category.

Response: Thank you for pointing out the typo. We revise Figure S14. Thanks!

Response to Reviewer #4

(1) I have joined the reviewing process following the withdrawal of Reviewer 1, and my comments are not intended to conduct a complete review of the paper but rather to assess to what extent the concerns raised by the initial Reviewer 1 have been appropriately addressed by the authors in their revision.

Reviewer 1 expressed reservations about the suitability of a $1^{\circ} \times 1^{\circ}$ resolution for the proposed analysis. I acknowledge the data availability constraints outlined by the authors and the specific objectives and limitations of the analysis presented. However, I believe that the authors need to more thoroughly discuss the implications of these limitations in the "Implications" section of the paper, where the main research findings are presented. The authors should provide deeper reflections on the constraints of the analysis and propose potential strategies to address them. At the moment, the "Implications" section is quite simple and does not provide enough discussion and context, similar to the degree of complexity and analyses provided by the paper. This section, as the most relevant section of the paper, should be more solid.

Response: Thank you for this suggestion. We add a separated paragraph in the "Implication" section (renamed as "Discussion") to address the caveat of coarse domain (Line 229-241). We clearly mention that a potential strategy to remedy the domain issue is to utilize a fine-resolution rainfall product, such as the 1-km MRMS rainfall dataset (only for the CONUS region). With a better product, a suite of advanced metrics that characterize fine-scale urban development patterns and a small domain can be incorporated. Thanks!

(2) Throughout the paper, the authors discuss the impact of urban form and development patterns on rainfall events, often referring to the relationship between cities and rainfall behavior. However, given the scale of the analysis ($1^{\circ} \times 1^{\circ}$), the findings are more applicable to large metropolitan areas and urban regions rather than individual cities. Therefore, I suggest that the authors revise the language throughout the paper to clearly indicate that they are working at an abstract scale, analyzing large urban regions and their aggregation patterns. The discussion should focus on debates about the agglomeration patterns of large urban regions rather than general debates

about the shape and form of cities. Including references to specific literature on the analysis of the shape and aggregation of large urban regions would enhance the paper's discussion (in opposition to debates about compact/sprawl city models, which refer to the analyses of urban shapes at detailed scales, which are not done in this case).

Response: We agree that the scale of analysis seems large for many small cities. The domain thus represents the aggregation pattern of several urban clusters (such as Zhoukou, China). For large metropolitan regions (such as Austin, US), the domain represents the shape of a city. We prefer to use the word 'city', assuming that either the aggregation pattern of small urban clusters or city shape is an attribute of a city. We made this clear in the revised manuscript (Line 89-90). Thanks!

(3) I think that the landscape shape index is enough to analyze the urban aggregation at the chosen scale. In this regard, authors are right when saying that, although many other metrics exist, they usually provide similar information and are very correlated. However, this is caused by the coarser scale of analysis, that hampers the characterization and measurement of urban aggregation pattern in a more accurate and precise way. Authors should acknowledge these limitations when discussing the results of the paper. As I explained above, it should be made clear, since the beginning, the objective and scale of the analysis that is carried out and its associated limitations (e.g., measuring urban aggregation at that scale can only be done through simple metrics such as the landscape shape index).

Response: We prefer to address this limitation in the Discussion section. This is because the choice of metrics is compromised by the analysis scale. Thanks all the same!

(4) In a similar vein, although the analysis does not directly account for the urban height, this is not very relevant because of the objectives and scale of the analysis that is done. Although future studies taking into account this factor, once global data datasets providing this information become available, would be interesting, authors should make clear that their objective and analysis is very limited and just provides a specific general understanding of rainfall interaction with the pattern of urban aggregation, an understanding and conclusions that could be further enhanced through complementary

and more detailed and ambitious studies.

Response: Done as suggested. Thanks!

(5) A similar comment can be made regarding the contrasting climates of cities and their effects on the analyses. The authors provide a supplementary analysis that proves how this factor does not directly affect the obtained results. Again, they should acknowledge that part of this conclusion comes from the objective and scales of analysis, comparing a vast number of large urban regions at 1°x1° resolution, so only very general conclusions can be obtained.

Response: We mention this limitation in Line 359-361. Thanks!

(6) All in all, the authors have effectively addressed the reviewer's comments. However, there remains a need for a more explicit acknowledgment of the main limitations identified by the reviewer within the paper and its discussion. The emphasis should not be on making extensive changes to the analyses conducted but rather on accurately framing the research question, explicitly outlining the limitations, and providing readers with a clear understanding of the study's contributions and the aspects that fall outside the scope of the analysis. This additional clarification will enhance the transparency and overall impact of the paper.

Response: Thank you for all the suggestions!